# Visual Discussion as Part of Internal Organization Communication—Functions and Significance

**Altti Näsi**

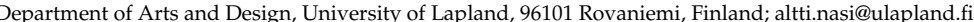

Department of Arts and Design, University of Lapland, 96101 Rovaniemi, Finland; altti.nasi@ulapland.fi

**Abstract:** Among an increasingly remote workforce due to COVID-19 pandemic, sharing photographs as part of internal communication has become something of a paradigm. In this article, exchanging primarily photographs and other quick visual artifacts, such as animated images and short videos, is considered a form of *visual discussion* among the work community. With a vast and diverse range of official and unofficial internal communication channels, this article focuses on three organizations, their internal communication channels and the visual discussions occurring therein. The semi-structured group interviews and qualitative thematic analysis we conducted shed light on the functions of photographs in different workplaces. The results demonstrate how visual discussions are heavily dependent on the context and nature of the work in question. In official channels, the most important functions of shared photographs are task-related and relevant to such issues as instructing, teaching, safety at work and emphasizing the message to be communicated. Photographs can also have a feeling-driven aspect that includes goals such as raising team spirit and employee commitment. Moreover, photographs are also shared in somewhat obscure unofficial channels with functions related to humour and concerning a common interest or hobby.

**Keywords:** visual discussion; internal communication; photography; workplace practices; photographic functions

## 1. Introduction

The digital revolution—or as some call it, the Fourth Industrial Revolution (Bonciu 2017)—has yielded new ways in which digital technology has become embedded in working life. This became topical during the COVID-19 pandemic with the need to overcome the challenges of mass telecommuting. For organizations, as consumers of new technology, a seemingly endless amount of software offers solutions in all business areas from financial management to human resources and from strategic thinking to operational activities, such as communication. This proliferation of technology provides the digital basis for almost all contemporary work industries and, as this article demonstrates, digitalization is not merely a technical question, but has altered our work-related modes of operation; tasks and working habits have changed and developed due to the pandemic, and in particular, in comparison to analogue times. This article focuses on one segment of this digital revolution: the communicative use of photographs, internally, in different working environments.

The aim of this data-driven study is to add to the understanding of the dynamics among an increasingly remote workforce and, more specifically, shed light on the functions of photographs shared by employees within internal communication channels. The photographs are used in a hybrid environment of official and unofficial channels.

As background, we know that internal communication articles underpin its relevance with organizational effectiveness (e.g., Quirke 2000; Welch and Jackson 2007; Tourish and Hargie 2004; Welch 2012; Ruck and Welch 2012). Internal corporate communication is typically aimed at promoting the commitment of the personnel to the organization, the awareness of its changing environment and a sense of belonging (also related to a sense of community). Effective internal communication strategies as a whole, and especially within

the work unit, are recognized as having a positive significance to the workforce (Therkelsen and Fiebich 2003; Saks 2006; Bindl and Parker 2010; Bakker et al. 2011; Mishra et al. 2014) and it also supports the external desired brand values (Aiello and Parry 2020, p. 236). As such, internal communication strives to take care of organizations' valuable assets; namely, the knowledge and interrelationships of its people.

From management and leadership perspectives, many organizations find commitment to the organization and a sense of belonging challenging with an increasingly remote workforce (e.g., Deloitte Global Human Capital Trends 2017). Corporate internal e-mails, newsletters, intranets and info screens have been in use for some time, but do not necessarily offer an effective way to inform, engage and reach staff immediately and efficiently. However, in reaction to an increasingly remote workforce and a networked environment, there are new means of communicating and interacting among individuals and groups in working life. This also includes the ubiquitous use of personal camera-phones and expanding communication to various new instant messaging applications, such as WhatsApp and Messenger.

In these circumstances, communicating with photographs is understandable, because the process of photography (the process of planning, shooting, editing and sharing photographs) has become virtually free of charge, as well as easier and quicker through digital networked cameras. It is safe to say that the ease of producing images and sharing them within the communities of which a person is part has become habitual. Modern community members share information, locations, advice, thoughts, feelings, aspirations, humor, goals, failures, successes, fears, dreams, likes and dislikes, to mention a few (Näsi 2020, p. 11).

The article at hand proceeds by first introducing the most significant terminology used. Then, it poses the research question and describes the methodology applied. Thereafter, the results and conclusion with mention of the theoretical and practical contribution of the study will be outlined.

## 2. Theoretical Background and Key Concepts

### 2.1. Internal Communication

As mentioned in the introduction, well-functioning internal communication is one of the major components of job satisfaction and efficiency and is often seen as a critical target of development. Without delving too deep into the revolutionary changes that have taken place during the last 30 years, concerning communication in general by reason of digitalization, some clarification is needed for the term *internal communication* when used in this study.

Typically, in scientific research, synonymous terms are used to describe similar issues and phenomena (see Hart 1998, p. 6; Hart 2001, p. 23). Concerning the nature of internal communication, the approach developed by Welch and Jackson (2007, p. 186) is seen as a suitable base because, according to them, internal corporate communication (ICC) is seen as a modern two-way environment in which managers and employees interact with one another (see also Wilcox et al. 2004; Van Riel and Fombrun 2007; Zerfass 2008). One of the major realizations is in recognizing the shift in focus from a one-way corporate communication instrument of management (a way to describe and explain organizational structure and define the communication content from top down) to a two-way approach, where communication is seen as multiway interactions between all groups and individuals as stakeholders. Relevantly, Welch and Jackson point out and identify the challenges of controllability and role of gatekeepers in regard to content in the digital era (Welch and Jackson 2007, p. 187). These ideas fit well with the idea that employees are not merely an audience, but possess and use a camera-phone and are themselves also producers and distributers of (visual) content.

Regarding the goals of internal communication, according to Welch and Jackson (2007, p. 188), they should consist of: (1) contributing to internal relationships characterized by employee commitment; (2) promoting a positive sense of belonging in employees; (3) developing their awareness of environmental change; and (4) developing their understanding

of the need for the organization to evolve its aims in response to, or in anticipation of, environmental change.

The four goals have a dependency on one another, particularly the first and second and the third and fourth (Welch and Jackson 2007, pp. 188–90). Ideally, contributing to internal relationships through communication means that the people working in an organization feel that they want to be there (also Baumeister and Leary 1995, p. 522; Paasivaara and Nikkilä 2010; Mishra et al. 2014). In other words, participation in internal communication could be seen as a manifestation of caring and meaningfulness.

Having control or making sense and organizing pre-planned and managed internal communication, and simultaneously taking into account informal chat on the "grapevine", can become challenging, as this paper later describes. Employees, temporary staff and subcontractors use social media and, as individuals, may become ambassadors or anti-ambassadors of the organization they work for (YLE 2022). As we know, there are cases where employees have felt mistreated and have published accusations towards their employers on social media. These types of cases demonstrate how defining internal and external communication (theoretically and in practice) and their audiences and participants, as well as the direction of communication and its content, is sometimes demanding.

Typically, the recipients of communication may seem easier to define, but controlling the further proliferation of the content is more difficult. Therefore, one could look at internal communication by applying Freeman's (1984, p. 25) stakeholder approach, which would mean that internal communication (photographic sharing included) incorporates all groups and individuals affected by the organization and recognizes them as partners in the dialogue (i.e., two-way communication). The use of camera-phones is just one common method of communication. Consequently, risk assessments concerning different communication channels become somewhat easier.

*2.2. Visual Discussion*

With the emergence of digital networks and the use of camera-phones with high-quality built-in cameras, non-professional photography is a significant form of everyday communication. Amateur photography has been studied since, at least, the 1970s (Sontag 1978; Chalfen 1987; Peters 2001; van Dijck 2008; Cobley and Haeffner 2009; Frosh and Pinchevski 2009; Lange 2011; Tait 2011; Sarvas and Frohlich 2011; Rose 2014; Gillies 2019; Zuromskis 2019; Näsi 2020). The preceding articles not only depict well the various functions of amateur photography, but also describe how it has evolved from analogue to digital.

However, the focus in earlier studies concerning amateur photography is almost without exception within leisure time and activities outside of work life. In this article, it is taken into account that amateur photography has inevitably become part of internal communication in most work organizations.

Therefore, the article at hand approaches photographic sharing from a specific point of view; the use of photographs as *visual discussion* (Näsi 2020, p. 101) within internal communication. The term *visual discussion* is used to describe the two-way visual communication that takes place primarily in the form of exchanging photographs within a work community. Indeed, visual discussion is typical inside any contemporary community, primarily in the form of exchanging photographs, but also other quick visual artefacts such as animated gifs, emoticons, emojis and short videos. This study focuses on the exchange of photographs. As in Näsi's work, other scholars such as van Dijck (2008), Van House (2011) and Villi (2014) have studied the revolutionary communicative changes in amateur photography. As the phenomenon is approached as a form of discussion, it could be compared to any other interactive form of communication that includes information and emotions (locations, activities, accomplishments, ideas, humour, likes and opinions) being shared among members of a particular community.

As noted, visual discussion is not merely restricted to work environments, but occurs in all areas of life and in one-to-one communication (see also Figure 1). Concerning the

earlier visual discussion studies outside working life, it is reported that the degree of image sharing tends to be more active among close friends and family than among members of other social groups (Näsi 2020, p. 12).

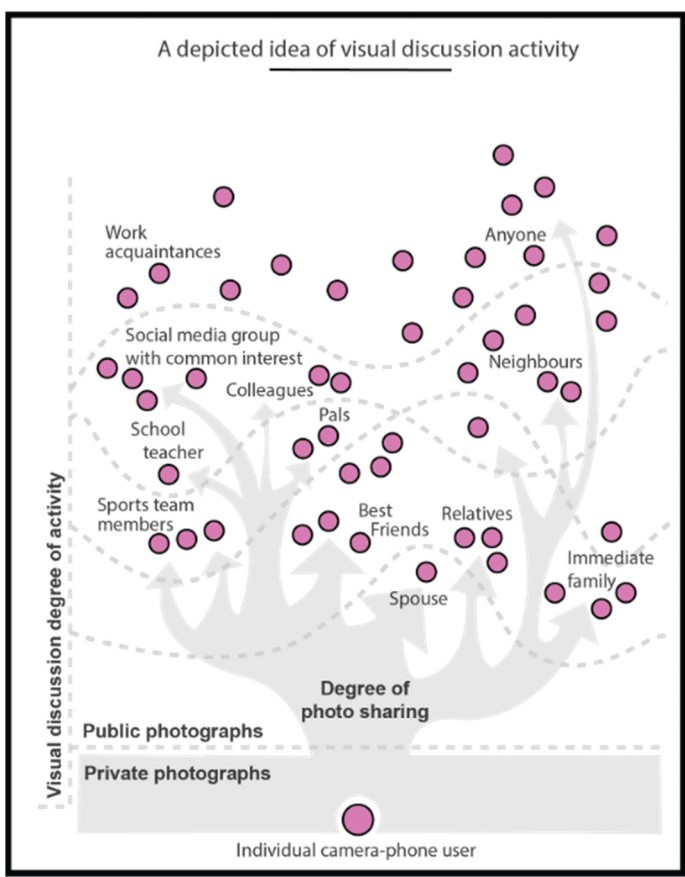

**Figure 1.** Outside the tasks defined by working life, the degree of visual discussions is most active among close friends and relatives (Näsi 2020, p. 12).

In the design stage of this study, it was assumed that visual discussion differs in work communities from that in communities outside of working life. This assumption was based on a notion that work-related visual discussion could oblige employees to take and share photos as part of different work assignments, whereas personal visual discussion is mostly dependent on personal choices and motivations.

### 3. Research Question

The aim of this article is to add to the understanding of the functions of camera-phone photography in various work communities. The research question is as follows: What types of visual discussions occur within various work organizations and what is their significance as part of internal communication?

### 4. Methodology

The research was data-driven and, thus, the results and contribution of this article are reached through qualitative research methods. The author conducted semi-structured group theme interviews, in person, on the premises of the three Finnish work organizations selected. Taking into account the limited number of organizations, the goal was to build a primary basis and a descriptive visual design to help increase understanding for future studies concerning workplace visual discussions. The organizations were chosen on an assumption that they would be as diverse as possible (educational establishment, banking institution and a construction company) and yet they would be representative organizations

of contemporary work life. Listed below are descriptions of the organizations involved, the underlined abbreviation used in this article, then the total number of workforce and the number of interviewees and their professional positions:

1. Adult Education Centre, AEC, ca. 250 employees, interviewed (3): Communications manager and two teachers.
2. Financial Group Company, FGC, ca. 11,000 employees, interviewed (2): head of communications and business partner manager.
3. Special Transport and Cranes Company, STCC, ca. 40 employees, interviewed (3): general manager, deputy general manager and branch manager.

The viewpoints of the results are management-oriented, as the chosen interviewees were chosen on the presumption that they have a more comprehensive understanding of the internal communication structures and solutions in their own workplace. In turn, the interviews were intentionally conducted as group interviews in light of a possibility for the interviewees to supplement each other's answers and, in this way, reach a more complete answer Hirsjärvi and Hurme (2015, p. 63). In the contacting phase and before conducting the interviews, the implications, goals or theoretical frameworks of *internal communication* and photographic sharing as *visual discussion* were not explained to the interviewees so that they would answer freely in the ways that they themselves comprehended and experienced these notions with regard to their own organization.

The semi-structured interviews comprised 15 questions and the structure of the interview applied Hirsjärvi and Hurme's (2015, p. 67) model, in which the themes are defined by the research question(s) at hand. Research Question Theme 1 (RQT1) establishes which channels are used for visual discussion, RQT2—4 add understanding to the functions of visual discussions and their significance to the organization and RQT5 reveals the views and experiences regarding the interface between work-related and leisure-oriented visual discussions. The number of questions on the topic are in parentheses.

RQT1: The acknowledged different official and unofficial channels for photographic sharing (2)
RQT2: The functions of photographs in these channels (2)
RQT3: The benefits and challenges of visual discussions (4)
RQT4: The significance of visual discussions to the organization (4)
RQT5: The interface between work-related and leisure-oriented visual discussions (3)

After the interviews, the given responses were transcribed and interpreted using thematic analysis (Guest et al. 2012; Eskola and Suoranta 2008, pp. 174–80). After transcription and printing, the different responses were carefully compared with each other in order to find patterns of themes. In practice, color labelling was used as coding to distinguish between similar, differing and unique responses (Guest et al. 2012, p. 64). Subsequently, the results were documented in Table 1, which is introduced in the following chapter. Thematically contradictory and unique data are introduced in the results chapter.

**Table 1.** The results obtained from the semi-structured group theme interviews and thematic analysis.

| Organization | Adult Education Centre (AEC) | Financial Group Company (FGC) | Special Transport and Cranes Company (STCC) |
|---|---|---|---|
| Official internal communication channels in use for visual communication | E-mail | E-mail | E-mail |
| | Microsoft Teams<br>Intranet | Microsoft Teams<br>Intranet | WhatsApp<br>Print |
| | Moodle<br>Futural Skills<br>Workseed<br>Kotopro | Yammer | |
| Similar and differing functions of visual discussion in official channels | - To raise team spirit<br>- Sharing successful work procedures | - To raise team spirit<br>- Sharing successful work procedures | - To raise team spirit<br>- Sharing successful work procedures |
| | - Colleague recognition<br>- Promote awareness of changing working environment | - Colleague recognition<br>- Promote awareness of changing working environment | - Instructions and (safety) information<br>- Monitoring/verifying/giving peace of mind |
| | - Monitoring the progress of studies<br>- Instructing, steering and reminding students<br>- To overcome language barriers | - To increase pride in one's work | - To increase pride in one's work |
| | | - Amplify the desired message<br>- Tool of management | |
| Unofficial internal communication channels in use for visual communication | WhatsApp | WhatsApp | WhatsApp |
| | Facebook | Facebook | |
| Similar and differing functions of visual discussion in unofficial channels | - Relates to a common interest during leisure time<br>- Minor controllability | - Relates to a common interest during leisure time<br>- Minor controllability | - An extension of the coffee room<br>- Builds a sense of community |
| | | - Highly humorous | - Highly humorous |
| | - Lacks consistency in content | | - Lacks consistency in content |

## 5. Results

Table 1 breaks down the responses of the interviewees as a result of the thematic analysis. At the top, there are the results concerning official channels and, at the bottom, those regarding unofficial channels. In the following sections, below Table 1, the results are introduced in greater detail alongside real-life examples.

The results in Table 1 show the first level data concerning the thematic analysis, particularly regarding RQT1, RQT2 and RQT3. For the interviewees, the organizations' official channels were easier to name and list than the unofficial channels because they had been created and were maintained due to recognized modes of using photographs at work. With unofficial channels, the knowledge and justification of their existence is more complex because they do not have the definite approval, or even necessarily the awareness of their relevance to work, among the organization's management. They have been typically created by the employees, and access and right to membership is not necessarily limited to individuals working in the organization. Moreover, the content shared in unofficial channels is not necessarily all work related. These findings put the notion of unofficial communication channels under critical scrutiny; if such a notion exists, what is the ownership of these channels and what are the connections with the work organization?

The differences between official and unofficial channels are salient, but the data obtained also emphasizes the importance of recognizing the need for strategic discussions on the functionality and organization of internal communication. Are unofficial channels created because of a demand not recognized by management? Only thereafter is it fruitful to discuss the different functions of visual discussions. Next, the similarities and differences emerging from the interviews are examined in more detail.

*5.1. Common Functions; Task-Related Photos to Share Information and Feeling-Driven Photos to Enhance Team Spirit*

The most obvious functions of visual discussions in the workplace deal with informing, instructing, facilitating and speeding up workflow. As an example, at the Special Transport and Cranes Company (STCC) it has become commonplace for the foreman of a project to inspect the worksite beforehand, take photographs and finally share the outcomes with the team responsible. Photographs are taken from areas and spots where special attention is needed. Heavy vehicles and machinery combined with construction and urban renewal sites involve occupational safety risks. The instructing photographs are habitually sent via WhatsApp, but often also printed on A4 paper and placed in the drivers' cabs of trucks, cranes and draggers. Hence, one of the major functions of visual discussion is in guiding and facilitating work processes by visual means. It also serves in taking care of employee safety and expensive equipment.

At the Adult Education Centre (AEC), visual discussions can be divided into two distinct categories: (1) sharing photographs among employees; and (2) using photographs as a method of teaching. Regarding the first category, raising team spirit, sharing successful work procedures, colleague recognition and promoting awareness of organizational changes were mentioned in the interview. Organizational changes included traditional announcements and bulletins from management to employees. One of the interviewees gave an example from the previous week of how the property manager had taken and shared a photo among employees of a filthy and messy sports hall locker room to remind teachers to pay attention to students neglecting common tidiness standards. The pictures also included some forgotten footgear and clothing en route to the lost property office.

As noted, the second category in regard to visual discussions at the AEC is the use of photographs as a method for teaching and learning. As the AEC is an educational institution for adults, a remarkably large number of its students are of foreign origin and struggle with the Finnish language. The AEC offers excellent vocational qualifications for many that have immigrated to Finland at an older age. The two interviewed teachers described how they use photographs taken by themselves to demonstrate, for example, how to operate a motor (photographs depicting where the buttons and various components are located and what they are called in Finnish) and what the key steps of a process to be taught are (photographs are serially numbered and work as bullet points). At the AEC, visual discussion is truly interactive; photos and small videos taken by students are utilized to demonstrate their acquired knowledge to teachers. Very commonly, teachers give assignments to students in which the task is to photograph or video record their operation and outcomes in exchange for study credits.

However, re-organizing and streamlining internal communication should be seen as a repetitive process, because visual discussions may also lead to overlapping communication channels. Among the employees at the AEC there had been some concerns about having too many channels, sometimes resulting in information (and image) overflow and therefore failed communication:

> "We do have a lot of communication channels. I still hear every now and then about someone not wanting to use a certain official channel like the Intranet, which results in staff not being absolutely on top of things. But of course, it's also the responsibility of the sender to make sure everyone receives and understands the content at hand. It's getting better, but it's definitely not perfect." (AEC, Communications Manager)

In all three of the organizations studied, visual discussion was also seen as a tool for building team spirit and enhancing employee commitment, regardless of the primary function of the content. This was considered particularly important at the STCC. There, sharing photographs was found easy and routine. This supports theories that the commitment aspect appears strongest in organizations where day-to-day communication occurs effortlessly, but simultaneously enhances workflow (e.g., Meyer and Allen 1997; De Ridder 2004; Reissner and Pagan 2013). For transport and construction workers, visual discussion was acknowledged as something that belongs to the job description and, as such, improves workflow efficiency and safety. Moreover, task-related photographs possess an emotional dimension. All three managerial interviewees at the STCC emphasized how photographs of completed projects and extremely challenging tasks are shared in the organization's official WhatsApp channel for the authentication and monitoring of the work progress between staff and management. Typically, an employee shares a photograph of a completed task in the WhatsApp-group with a caption "Mission accomplished" or something similar.

Soon after the completion of a project, some of these photographs are frequently printed as posters and hung on the walls of the workplace. Finally, at the end of the year, some of the best photographs end up in the following year's wall calendar given to employees and other stakeholders. The calendar includes 11 pictures of the most significant achievements of the previous year. The 12th photograph is typically a black and white picture chosen from an historical work mission from years ago in order to acknowledge the traditions of the company.

> "These pictures give a feeling of commitment, satisfaction and pride. It's when we've managed to complete a challenging task, it generates lots of pride in our work community and the photographs embody that feeling. Afterwards, it's nice to look at these photos as they are displayed at the office." (STCC, General Manager)

Similarly, at the Financial Group Company (FGC), promoting a positive sense of belonging in employees is encapsulated by calling it also *a we-feeling* or as a feeling of a common tribe. This is often achieved conveniently by using photographs and small videos, which are also considered to save time at work and emphasize the message to be conveyed.

> "Our workflow is quite demanding and sometimes it's hard to find the time to read long texts. Photographs may offer a kind of a shortcut. They may save time and we think that they emphasize the desired message . . . we occasionally also use professional photographs in internal communication because we don't want our visual identity to vary too much." (FGC, Communications Manager)

In further discussions, it was mentioned that the dynamics of using amateur versus professional photographs from a time consumption viewpoint can sometimes become complex. For example, there are similarities but also difference in the outcomes of sharing amateur snapshots with colleagues from an FGC internal mass event or of hiring a photographer to take equivalent promotional photographs. Both photographs would be taken from the same event and depict what happened, but amateur photographs may elicit a more straightforward evidence-like reaction to what happened. The interviewees agreed that visual discussion originating from employees has a different perspective from that of a professional. In her research (Salo 2002, p. 108), Salo describes it as a window to reality and authenticity. Puustinen and Seppänen (2011, p. 189) have also demonstrated that amateur images in the media are often considered truthful and even more trustworthy than photos taken by a professional photographer. In both studies, rough, imperfect images taken by amateurs provide immediate and authentic testimony to the fact that the bystander has been on the spot. In contrast, pre-planned and mission-driven, skillfully edited images may impair this authenticity and spontaneity, despite possibly successfully maintaining the desired corporate visual identity (CVI) according to the instructions.

The FGC interviewees summarized the functions of shared photographs under three categories: (1) they are capable of adding truth value; (2) they render a message more

interesting; and (3) they can make complex content easier to understand. Regarding the truth aspect, value and weight were finally linked to an idea of having a recognizable person from management alongside the desired message. A key figure was thought to imbue the task-related messages with greater noteworthiness and credibility.

Proceeding with FGC's thoughts on a shared tribe and raising team spirit, with over 11,000 people working for the organization, identifying colleagues nation-wide is understandably challenging. Adding many kinds of visual elements such as passport-like photos to all personal user profiles (intranet, ID-cards, e-mails and social media) creates a more personal connection between colleagues and is considered to increase team spirit. According to the interviewees, photographs of colleagues are regarded as helpful in lowering thresholds when contacting each other, as well as building a sense of trust and closeness.

In contrast to smaller organizations, the use of facial photographs alongside user profiles was mentioned only by the large-scale corporation, FGC. At the STCC, with only 40 employees, management assumes and requires that everyone knows each other. There, using facial photos or including pictures of directors as part of internal communication was considered unnecessary and even ridiculous in light of their needs. Therefore, it is important to understand that photographs may be considered as elements of promoting team spirit in a multitude of ways but that the organization's size and industrial sector has a decisive effect on the functions of visual discussions.

The detailed ways in which photographs were used entailed acknowledging the differences not only between organizations, but also within an organization, as described above. This is in accordance with Welch and Jackson (2007, p. 186), who recommended not seeing the personnel as a uni-dimensional single public. Considering the results in this study, the way in which visual discussion is led and participated in depends on the work culture and the job description of an employee.

Developing the awareness of environmental change and the understanding of the need for the organization to adapt its aims in response to environmental change was most recognizable at the FGC. The Head of Communications related to visual communication as a tool for explaining, clarifying and justifying necessary future changes no matter how large. Visual discussion was not only able to emphasize the desired message, but including photographs of management was seen to lend credibility and significance to communication. As for the STCC and AEC, the viewpoints were more practical in the sense that the benefits of using photographs were described as mostly task-related and concerning concrete actions.

Finally, it is also important to understand that visual discussions may push through very comprehensively and make long-lasting impressions. For example, at the STCC, photographs of completed tasks and worksites are displayed on the office walls and in the calendar for the following year, as we have learned. Hence, their primary function evolves through time. Surely these photographs also have an impact on employees and immediate stakeholders or anyone visiting the STCC lunchroom or conference hall. The pictures on the wall and the piles of old calendars reveal the mental landscape of the company's management and their urge to visually build team spirit among employees.

### 5.2. The Functions of Unofficial Channels

During the interviews, it became clear that the belief in the appropriate use of photographs rests heavily on trust. None of the three organizations had written instructions on how employees could specifically use photographs in their work. The complexity and problem of determining the borders between internal and external communication were also perceived. The Financial Group Company (FGC) and the Adult Education Centre (AEC) have social media guidelines but there are no detailed guidelines on the use of photographs. All of the interviewees called for responsibility on the part of employees and the use of common sense. Common sense was explained as a putative understanding among the employees to use consideration and self-criticism when taking and sharing

images; causing harm to the workplace causes harm to oneself and, thus, gatekeeping was seen as something of an automated process.

However, all three organizations had faced some minor issues due to a lack of consideration regarding photo sharing, but not to the extent that using a camera-phone at work would necessitate separate instructions, at least so far. The issues in question had been dealt with by having a discussion with the employee concerned about the inappropriateness of the image(s) and the required change in behaviour.

Regarding the viewpoint of unofficial channels and raising team spirit among employees, at the FGC, there were unofficial channels in which photographs were shared to promote mutual interests, the content often being associated with in-group humor that would be difficult for outsiders to understand.

> "What I know of some of the communication in the unofficial channels is that humour plays a significant role. Much of the content may be *inside humour* that wouldn't be understood by people not working for us or in close relation to someone working for us." (FGC, Communications Manager)

As an example, employees interested in jogging have started a Facebook group to bring together like-minded people and to make running together easier by planning group runs and sharing pictures, information and thoughts about jogging and a healthy lifestyle. With company-coloured t-shirts and a health-promoting hobby, the primary positive function of the visual discussions is seen to be to promote a company-related tribe feeling beyond working life. It was said that these photographs were rarely, although sometimes, used more extensively to display togetherness and employee well-being in internal PowerPoint presentations, for example. In this regard, it is important to note that this type of visual discussion could also have a counter-effect on those not involved in the running club because it could create a feeling of exclusion and inadequacy (Paasivaara and Nikkilä 2010, p. 49). A sense of community comprises a notion of "us" and "them" and, in this case, the running club divided employees into those who participated and those who did not participate. Using such images to promote team spirit and a sense of community would obviously be ill-advised.

Lastly, an unofficial channel may also serve as an extension of the coffee room, as it was described at the STCC. There, the purpose of this unofficial WhatsApp group was to include everyone in the daily chitchat, despite not necessarily being physically present. The content was highly humorous, and its main function was to maintain a sense of community by means of photographs, short videos, gif animations and witty comments. It was also noted that the content typically made no or very little reference to work.

## 6. Conclusions

### 6.1. Theoretical Contribution

In accordance with Näsi's (2020, p. 101) concept, visual discussion refers to communication occurring inside any contemporary community, primarily in the form of exchanging photographs. These photos are tools for discussion, through which, for example, locations, activities, achievements, ideas, humor, likes and opinions are shared among other members. However, visual discussion in work communities has distinct nuances.

One of the major theoretical contributions of this article is to divide the function of visual discussion in working life into two main categories: (1) task-related; and (2) feeling-driven. During the life cycle of an individual photograph, its main function may shift from one category to another.

When concerning the theoretical frameworks (Therkelsen and Fiebich 2003; Saks 2006; Welch and Jackson 2007; Bindl and Parker 2010; Bakker et al. 2011; Mishra et al. 2014) referenced at the beginning of the article in relation to visual discussion, photographs are able to contribute to employee commitment and promote a positive sense of belonging due to their manifold options for use. In fact, visual discussion works cross-sectionally on two levels. Firstly, and unsurprisingly, consciously chosen shared photographs of successfully completed work tasks or humorous memes, for example, contribute to promoting a positive

sense of belonging, thereby raising team spirit. Well considered responses build a positive sense of community at work. As another example, including images of management alongside visually depicted future goals and aims set may increase the weight of the message, as mentioned at the Financial Group Company (FGC). There, the photographs were most consciously used to explain internal and external changes in the environment, which is partly due to an exact reporting culture, in which management issues official yearly, quarterly and monthly announcements. Visuals are harnessed to explain goals for the immediate future.

On a second level, it is important to take into consideration that the way in which visual discussion is harnessed represents the mindset of management. For example, promoting successful achievements and acknowledging the employees involved is a conscious act. The interviews revealed how pictures of challenging but successfully completed construction jobs at the Special Transport and Cranes Company (STCC) are first shared in the organization's official WhatsApp group, then some of the photos are printed and fixed on the office walls and finally some of the pictures end up in the next year's wall calendar. Hence, the functions of visual discussion portray management's pride in the employees and their achievements. The act of prominent photographic sharing could be seen as a tool for stating company values that the work really matters and that the long-term traditions and history are well cherished and appreciated. After all, managing and browsing through photographs, printing and hanging them up on the wall and making a yearly calendar all consume time and resources. These results are compatible with the research by Reissner and Pagan (2013), for example, where employees respond positively to communication that makes them feel valued and involved. It enhances their propensity to engage with the organization.

In light of the results summarized in Table 1, the task-related functions of visual discussions are sometimes given less attention in theory-based concepts. For example, some of the discovered task-related photographs would be challenging to incorporate with the four goals of internal communication by Welch and Jackson (2007, p. 188); particularly at the Special Transport and Cranes Company (STCC) and the Adult Education Centre (AEC), where visual discussion had direct practical functions contributing to work safety, workflow, cost efficiency, teaching, to remove language barriers, to give study assignments and to demonstrate student knowhow. In a nutshell, visual discussion contributes by facilitating planning, execution and displaying the final outcomes and therefore these primary functions should be dissociated from the later implications that follow (see also Figure 2).

When acknowledging the more strategic standpoints of various scholars working with internal communication, it is nevertheless suggested that when analysing visual discussions in an organization, the proximity of goals and content should be recognized. Furthermore, the content of visual discussion may create new goals, at least concerning the functions where shared photographs make working life smoother and more effective. Of course, this requires overall scrutiny of where and how visual discussions happen.

Finally, as a theoretical contribution of this article, it demonstrates that the so-called official photographs in internal communications function notably differently when compared to the family and friend photography depicted in Figure 1. Some of the functions in a working environment do not exist concerning leisure time (for example, recognising a colleague out of a thousand others) or they may have minor significance (for example, promoting the awareness of changing working methods). Most importantly, outside working life, personal visual discussions are driven by inner motivation and are not tied to job description or work tasks. However, some of the differences apply to different-sized workplaces; the role of visual discussions was very different at the Financial Group Company (FGC), with 11,000 employees, from that at the Special Transport and Cranes Company (STCC), with around 40 employees. As a family business with fewer workers, visual discussions could more closely resemble that within a family.

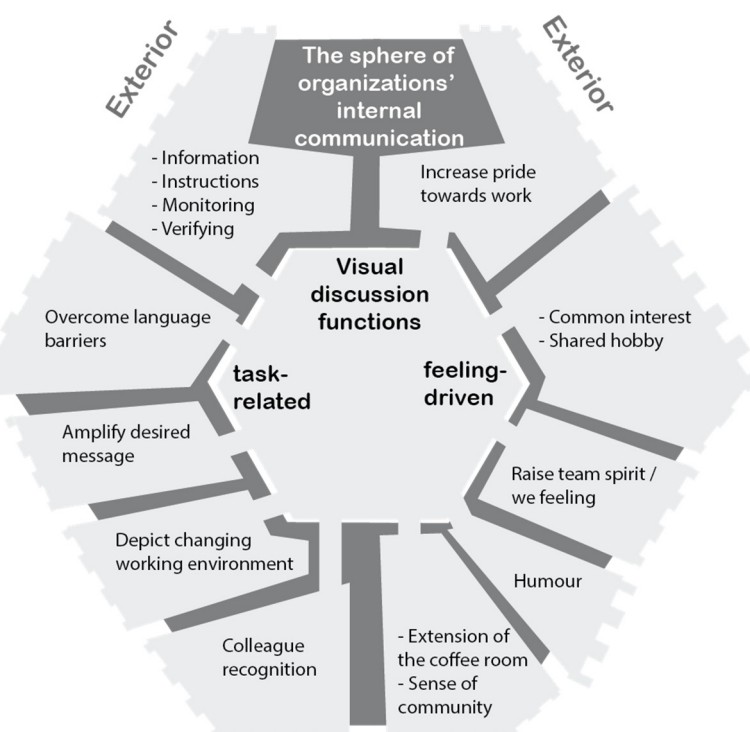

**Figure 2.** A summarized visual representation of the functions of visual discussions as part of internal communication in the studied organizations.

*6.2. Practical Contribution*

The practical contribution of this article could be divided into four main categories. Firstly, we take the view that an organization's official photographic sharing is perceived as something that should be under the supervision of management, and simultaneously the challenges of gatekeeping are well acknowledged due to the ubiquitous presence of camera-phones. Furthermore, it is accepted that the functions of sharing photographs are non-traditional in regard to a controlled and consistent corporate visual identity (CVI). The gatekeeping process of a visual identity becomes most challenging when the whole workforce is engaged in photography and sharing the results, as described by the communications management of the Financial Group Company (FGC).

The challenges of gatekeeping have increased due to experiences of failed photo sharing (intentionally or accidentally). As an example, from this article, the Special Transport and Cranes Company (STCC) used snapshots as tools for instructing work tasks. However, it was acknowledged that these same photographs include an enormous amount of risk factors, such as reputational risk in case of an accident or operational error. In addition, revealing important business secrets (regarding partnerships, customers and working methods) was mentioned as a risk of failed photo sharing. In the organizations involved in the research, photographic sharing among the workforce is currently mainly based on trust. Management believes that employees comprehend how ill-considered photo sharing is similar to shooting oneself in the foot; it would affect their own workplace and consequently their own well-being. However, a practical contribution or recommendation of this paper is to point out that media literacy and, more specifically, guidance in visual discussions is recommended, due to the risk of unintentional blunders. Furthermore, in case of disagreement or failure to be discreet, employees could use the material from visual discussions to damage a current or a former employer.

Secondly, based on the findings of this article, visual discussion should be seen as a strategic management tool and a natural part of internal communication. The benefits of presenting complex information in visual ways is comprehended; photographs are able to amplify the desired message, inspire employees, raise team spirit and increase a healthy

pride in the work itself, to mention just a few. With the Adult Education Centre (AEC) and the Financial Group Company (FGC), requiring everyone to attach a facial photograph to messages as part of internal communication is a conscious decision made by management. According to the results, using recognizable faces, attaching facial photographs and using visual discussion to distinguish and familiarize individuals, make internal communications more intimate and thus more meaningful.

Photographs can explain visually: "this is who we are and this is what we do". Critically speaking, although all three organizations considered visual discussions to be something that improves a sense of community among the workforce, they simultaneously struggled to name the concrete factors underlying this phenomenon or the studies that the notion was based on. For organizational management (as well as for future studies), it would seem sensible to study the themes that strengthen and weaken team spirit and how photographic sharing fits into this scheme.

Thirdly, the role of unofficial channels at workplaces is indefinite to the extent that they are often hard to determine as belonging to the organization and their relevance to work tasks can also be challenged. This does not mean that their functions are merely feeling-driven, because they may be fixed around a shared interest, such as running, and may contain information, advice and recommendations. In the case of the running club at the FGC, its main function was to bring together like-minded people and to make running together easier. However, an unofficial channel may also serve as an extension of the coffee room, where everyone can participate, despite being physically apart, as at the STCC. It should be taken into consideration that the content tends to be highly humorous by means of sharing photographs. Analysing the role of other pictorial elements, such as short videos, gif-animations, emoticons, emojis and comments alongside photographs provide a fruitful basis for further research.

If we think about ownership responsibility, copyright issues, and publishing permissions, freedom of speech and other privacy policies, some formal agreement of a written resignation by the management concerning photographic sharing in unofficial channels is recommended. Of course, this would be difficult if unofficial channels are not acknowledged. Furthermore, making a negative statement about unofficial channels raises questions regarding their demand and reason for existence in the first place. This may also entail reorganizing or extending official internal communications in some way. With the Financial Group Company (FGC) the options and guidelines on using photographs were the most advanced, although the emphasis was clearly on external communications and developing their own CVI.

Finally, the idea of being mobile and available while on the go has, in many organizations, come to be taken for granted. In contrast to the times when one had the option of calling another person at the office (non-mobile) or while away from the office (mobile), the telecommuting development has accelerated to a whole new level due to the on-going COVID-19 pandemic.

Organizations can fortunately develop and profit from well-managed visual communications. In summary, the personal camera-phone serves as a practical tool for immediate two-way communication within organizations. Telecommuting is one of the major reasons that has forced—or, one could say, steered—organizations to look at internal communications predominantly as a two-way or even multi-way process on a whole new level. Work-based software, such as Microsoft Teams, alongside many social media platforms should also be seen from a democratization and participative viewpoint, providing everyone with a voice and a direct channel for discussions. It is suggested that visual discussions are yet another reason why organizations must be prepared to react within the existing two-way communication channels, particularly in cases of challenges and conflicts as they occur. As we know, the right kind of rapid reaction to an incipient crisis often minimizes reputational damage.

The results of this study help add to the understanding of how shared photographs function as part of internal communications in different work organizations. It not only

succeeds in revealing the wide range of functions that visual discussions encompass, but also the life cycle of the shared photographs, because the functions of a specific photograph may evolve in the course of time.

The adaption and ease of incorporating photographic sharing into almost all modern communities make visual discussions special. In fact, exchanging photographs via cameraphones is a mode of internal communication that has, in a way, infiltrated corporations. Due to an increasingly remote workforce, organizations have had to adapt to new methods of communication.

**Funding:** The Finnish Work Environment Fund (Työsuojelurahasto), Fund #210001.

**Institutional Review Board Statement:** Ethical review and approval were waived for this study due to maintained privacy and anonymity regarding organizations and individuals throughout this article.

**Informed Consent Statement:** Informed consent was obtained from all subjects involved in the study.

**Data Availability Statement:** The data presented in this study are available on request from the author.

**Conflicts of Interest:** The author declares no conflict of interest.

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
