# Peer review of "Visual Discussion as Part of Internal Organization Communication—Functions and Significance"

_journalmedia, doi:10.3390/journalmedia4010005_

Round 1

Reviewer 1 Report

The paper is well-written and original but relies heavily on this source concerning the central term "visual discussion" by Näsi, 2020.

this paper is an interesting interdisciplinary mixture of visual studies and public relations. The author might benefit from quoting also "Visual Communication " by Aiello, Giorgia; Parry, Katy 2020. Recommended source: https://icavisualcommunicationstudies.com/about/ica-vcs-members-bookshelf/    Sum up- Good paper.

Author Response

Dear Reviewer,

Your comments are of great value and I have made improvements according to your suggestions, thank you. Furthermore, your encouraging comments are appreciated.

- The notion of "Visual discussion" by Näsi is indeed the other central term alongside "Internal Corporate Communication" by Welch and Jackson. The two theoretical background concepts attempt to add to the understanding of photographic exchange within different organizations. I have made highlights to the other supporting theories that are inline with the two key concepts.

The book "Visual Communication, Understanding Images in the Media Culture" by Aiello Giorgia and Parry Katy was found very interesting and helpful as well. I have added a quote suitable for this article from the book and added it to the references.  

Best Regards!

Reviewer 2 Report

- In creating an outline of the study, other pictorial materials should have been included

Infographics, illustrations, symbols and signs (emoticons, emojis, etc.) and all other pictorial elements are an important part of visual communication, and I assume that they are used a lot in informal communication within various organizations.

- If analysis and comparison of pictorial elements were included, as part of the quantitative method in research, the study would be more original 

Author Response

Dear Reviewer,

Your comments are of great value and I have made improvements according to your suggestions, thank you. Furthermore, your encouraging comments are appreciated.

With respect, I interpret all three of the Reviewer's comments to relate with the issue of taking into consideration also other pictorial elements in the text and in the conclusions sections. Unsurprisingly, to add other pictorial elements was noted and taken into consideration already when designing this study. Unfortunately, it was seen as a topic that would deserve a study of its own that could and should have liaisons to this study and others similar. It is admitted, that performing study with these other pictorial elements would produce highly original material.

Based on the Reviewer's point of view, other pictorial elements have been added as important elements to consider when talking about modern internal communication (article: Visual Discussion and Conclusion sections).

Best Regards.